# Applications of Genome Editing Technologies in CAD Research and Therapy with a Focus on Atherosclerosis

**DOI:** 10.3390/ijms241814057

**Published:** 2023-09-13

**Authors:** Michelle C. E. Mak, Rijan Gurung, Roger S. Y. Foo

**Affiliations:** Cardiovascular Research Institute, Cardiovascular and Metabolic Disease Translational Research Programme, Yong Loo Lin School of Medicine, National University of Singapore, 14 Medical Drive, MD6, #08-01, Singapore 117599, Singapore; michelle.m@u.nus.edu (M.C.E.M.); roger.foo@nus.edu.sg (R.S.Y.F.)

**Keywords:** atherosclerosis, coronary artery disease, genome editing, CRISPR, ZFN, TALEN

## Abstract

Cardiovascular diseases, particularly coronary artery disease (CAD), remain the leading cause of death worldwide in recent years, with myocardial infarction (MI) being the most common form of CAD. Atherosclerosis has been highlighted as one of the drivers of CAD, and much research has been carried out to understand and treat this disease. However, there remains much to be better understood and developed in treating this disease. Genome editing technologies have been widely used to establish models of disease as well as to treat various genetic disorders at their root. In this review, we aim to highlight the various ways genome editing technologies can be applied to establish models of atherosclerosis, as well as their therapeutic roles in both atherosclerosis and the clinical implications of CAD.

## 1. Introduction

Despite advances in medicine, cardiovascular diseases (CVDs) remain one of the leading causes of death worldwide. As of 2019, CVDs account for almost one-third of deaths globally [1,2] with cases increasing significantly since 1990 [3]. CVD is an umbrella term for a variety of diseases that affect the heart such as stroke, coronary artery disease, and hypertensive heart disease [4,5]. Often, these diseases can result in heart failure (HF), with coronary artery diseases (CADs), in particular, being indicated as one of the leading forms of CVD related mortality and disease burden, contributing to 49.2% of all CVD related deaths [3,4,5].

CAD mainly results from plaque formation in the intima of arteries, resulting in the narrowing and eventual occlusion of the vessel, a disease known as atherosclerosis [6]. In the clinic, CADs manifest mainly as myocardial infarctions (MIs) or ischemic cardiomyopathies [7]. While reperfusion is often the main therapy for instances of MIs, they may sometimes result in ischemia/reperfusion injury, leading to myocardial dysfunction and eventual heart failure [8]. It is, thus, unsurprising that the study of atherosclerosis is highly important in not only the understanding but also the potential treatment of CAD.

Much effort has been made to understand the genetic risk factors underlying CAD and, consequently, potential areas for intervention. CAD is a complex disease influenced by both the environment and genetics. In recent years, advances in sequencing technologies have allowed for the identification of various gene loci that are significantly associated with CAD [9]. Genome-wide association studies (GWAS) on various cohorts have identified various genetic risk loci that are associated with CAD. Many of the genetic loci identified lie near genes with roles in the metabolism of cholesterol and lipoproteins such as *PCSK9*, *APOB*, and *ANGPTL4* [10].

Many population studies on atherosclerosis have also identified elevation of low-density lipoprotein (LDL) cholesterol and apolipoprotein B (APOB) 100 to be associated with risk of atherosclerosis related cardiovascular events. In particular, LDL cholesterol has been identified as a causal factor in the development of atherosclerotic cardiovascular disease [11]. GWAS studies have further identified several risk loci associated with both CAD and MI [12,13] in genes known to affect plasma LDL levels. Conditions such as familial hypercholesterolemia have been linked to increased cholesterol levels and increased risk for CAD. This condition is characterised by mutations in *LDLR*, *PCSK9*, and *APOB*. Unsurprisingly, many therapeutics aimed at managing CAD have also been aimed at lowering the levels of LDL cholesterol (e.g., statins).

In this review, we will briefly cover the pathophysiology of CAD, and how genome editing technologies such as CRISPR can assist in the further understanding of and treatment for CAD.

## 2. Pathophysiology of Atherosclerosis

Atherosclerosis is an inflammatory disease [14,15] characterised by the formation of plaques that result from the accumulation of lipids and inflammation of the arteries. It can clinically manifest with either acute or chronic presentations in a variety of ways, depending on the vascular territory involved, and have both acute and chronic presentations [16]. For example, plaques that form in the coronary arteries result in CAD, while those that form in the legs and/or arms result in peripheral artery disease (PAD), and those that form in the carotid and cerebral arteries can present as ischemic stroke [17]. Low-density lipoprotein (LDL), in particular, has been implicated as having a causal effect on the initiation and progression of atherosclerosis [11], though it is still unclear how exactly excessive LDL cholesterol results in atherosclerosis. The accumulation of LDL and other lipoproteins in the intima of arteries activates the endothelium and, in turn, triggers a series of events that lead to inflammation and eventual uptake of modified LDL cholesterol by macrophages, forming what is known as “foam cells”. Over time, these plaques continue to grow through the accumulation of lipids and foam cells derived from both macrophages and smooth muscle cells (SMCs) [18,19]. Impaired efferocytosis due to plaque development eventually leads to an accumulation of apoptotic cells in the plaque, driving the formation of a necrotic core [20]. This, in turn, results in the formation of a fibrous cap and calcification.

Eventually, “vulnerable” plaques rupture, exposing the necrotic core to the circulation. This, in turn, results in the activation of tissue factor [21], which subsequently triggers a cascade of events such as the recruitment of platelets and inflammatory cells, eventually forming a thrombus. Occlusion of a vessel by a thrombus eventually leads to ischemia (low oxygen) and ultimately infarct (tissue death). If these occur in the coronary arteries, they could result in clinical manifestations of myocardial infarction and angina. If a thrombus occludes cerebral arteries, it could result in a stroke [22,23]. On the other hand, a thrombus may dislodge and block a distal vessel elsewhere in a process known as thromboembolism, which could lead to ischemia and potential infarction of neighbouring tissue.

Lipid-lowering drugs have been the primary therapeutic strategy for managing atherosclerosis. Statins, 3-hydroxy-3-methylglutaryl-CoA reductase (HMG-CoAR) inhibitors, have long been used to lower LDL cholesterol levels, and large meta-analyses of statin trials have shown that more intensive statin therapy was significantly associated with further reductions in major vascular events. More importantly, these trials showed that reductions in LDL cholesterol levels result in reductions in the rates of major vascular events such as strokes and nonfatal myocardial infarctions [24]. Nonstatin lipid-lowering drugs such as Ezetimibe have also been used in combination with statins to further reduce LDL cholesterol levels by 15–20% [25].

On the other hand, the discovery and understanding of the function of proprotein convertase subtilisin/kexin type 9 (PCSK9) enzyme in lipid metabolism has led to the development of PCSK9 inhibitors to treat hypercholesterolemia and atherosclerotic cardiovascular disease. PCSK9 regulates cholesterol homeostasis by binding to the LDL receptor (LDLR) and promoting its degradation [26,27]. Loss of function mutations in PCSK9 have been linked to reduced levels of LDL cholesterol and lower incidences of coronary heart disease [28,29,30,31,32]. On the other hand, gain of function mutations such as the PCSK9-rs562556 variant in the STANISLAS cohort [33] have been linked to higher PCSK9 levels and are associated with carotid arterial plaques. PCSK9 inhibitors such as evolocumab [34,35,36] and alirocumab [37] have been evaluated in clinical trials and have shown significant efficacy in reducing LDL cholesterol levels and incidences of cardiovascular events. Additionally, nucleic-acid-based therapies such as inclisiran, a small interfering RNA (siRNA) therapeutic targeting PCSK9 mRNA, have also been shown to be effective in lowering PCSK9 and LDL cholesterol levels in patients with elevated LDL cholesterol levels [38,39].

## 3. Genome Editing Approaches

In recent years, many gene editing systems have been identified and developed for various purposes. Gene editors not only allow us to understand disease mechanisms and gene function, but also harbour potential for therapeutic applications. However, issues of safety and efficiency must be answered before widespread adaptation. A summary of the following approaches and their pros and cons can be found in Table 1.

### 3.1. Zinc-Finger Nucleases

Zinc-finger nucleases (ZFNs) are chimeric restriction endonucleases comprising a DNA binding zinc-finger protein domain and a *Fok*I catalytic domain [40]. The dimerization of the *FokI* catalytic domain is necessary for its function; thus, two sets of zinc fingers must bind to their target sequences in close enough proximity with the correct orientations [41]. Upon successful binding, DNA cleavage occurs, generating a double-stranded break (DSB) at the target site. DSBs can be repaired by nonhomologous end joining (NHEJ) or by homology-directed repair (HDR) (Figure 1).

The zinc-finger protein domain contains a tandem array of Cys2-His2 fingers that recognizes approximately 3 bp each. Depending on the target, the number of fingers per zinc-finger protein can range from three to six, corresponding to a recognition site of approximately 9–18 bp. ZFNs have been successfully used to generate knockout cell lines such as a *DHFR*^−/−^ Chinese hamster ovary (CHO) cell line [42], as well as insert corrective transgenes [43].

Additionally, ZFNs have also been utilised to generate animal models to further the understanding of diseases. For instance, Yan et al. utilised ZFN-mediated gene editing to specifically knockout apolipoprotein C-III (apoCIII) in rabbits to understand its role in atherosclerosis [44]. Similarly, LDLR knockout rats [45,46] have been generated to provide animal models of hypercholesterolemia and atherosclerosis.

### 3.2. Transcription-Activator-like Effector Nucleases

Similar to ZFNs, transcription-activator-like effector nucleases (TALENs) carry out their genome editing activity by recognition and cleavage of the target sequence via a double-stranded break. These nucleases consist of transcription-activator-like effectors (TALEs), derived from the plant bacteria in the genus Xanthomonas, linked to a *FokI* catalytic domain. TALEs contain a central repeat domain that allows for DNA recognition, with a repeat unit of 33–35 amino acids specifying one target base [47]. DNA recognition specificity is encoded by the highly variable amino acids at positions 12 and 13, referred to as the “repeat variable di-residue” (RVD) [48]. Much like ZFNs, a pair of TALENs is required for the successful cleavage of the target site, generating a double-stranded break.

The foundations of these TALEN modifications were first laid out in the 2011 study by Miller et al. [49]. Since then, multiple modifications have been made to TALENs to enable greater gene editing efficiency as well as to improve their specificity. Numerous methods have also been developed to enable cost-effective assembly of TALE repeats, allowing for efficient production [50,51,52]. While TALEN-mediated gene editing is often characterised as being less efficient than CRISPR, TALENs have the benefit of inducing less off-target editing [53] as well as a greater editing efficiency in heterochromatin target sites [54]. Notably, a TALEN-based knockout library has been utilized to develop human-induced pluripotent stem cell-based (hIPSC) models of cardiovascular disease. Karakikes et al. designed a collection of TALENs that knocked out 88 human genes associated with cardiovascular diseases by inducing induced double-stranded breaks near their associated start codon [55]. More recently, preclinical data from Cellectis showed successful gene editing of *HBB*, a gene linked to sickle cell disease, utilising TALENs [56].

### 3.3. Clustered Regularly Interspaced Short Palindromic Repeats (CRISPR) and CRISPR-Associated (Cas) Systems

CRISPR was first discovered in *E. coli* and other bacterial species [57]. In bacteria and archaea, the CRISPR/Cas system acts as a defence and regulatory system to protect the host from external insult [58,59,60,61] via the integration of the foreign DNA into the host CRISPR locus. Unlike ZFNs and TALENs which rely on protein–DNA interactions for targeting, Cas nucleases rely on short RNA sequences (termed guides) that recognise the target DNA sequence.

The most commonly used Cas protein derives from *Streptococcus pyogenes* (SpCas9), a type II CRISPR/Cas system. Unlike type I and type III CRISPR/Cas systems, type II systems rely on a single Cas protein for DNA cleavage [59]. In type II CRISPR systems, the CRISPR locus is transcribed to produce pre-CRISPR RNA (crRNA). Concurrently, trans-activating RNA (tracrRNA), involved in processing of pre-crRNA to crRNA [62], is also transcribed from a region upstream of the CRISPR locus. Seminal work by Jinek et al. found that both tracrRNA and crRNA are required for the cleavage of target DNA. Additionally, they found that it was possible to target the Cas9 endonuclease to specific DNA sequences by engineering guide RNAs (gRNAs) or single guide RNAs (sgRNAs) that contained both the target recognition sequence and the tracrRNA [63].

However, there are limitations in the regions that CRISPR/Cas systems can target for editing. The Cas nuclease first looks for its cognate protospacer adjacent motif (PAM) before assessing guide–target complementarity. Thus, targetable sequences in the genome are limited by the presence of a PAM motif downstream. Fortunately, there are a wide range of PAM motifs recognised by different Cas nucleases, and efforts have been made to characterise and engineer Cas nucleases to expand the range of targetable sequences [64,65,66].

Following successful targeting of the Cas nuclease by gRNA(s), the nuclease creates a DSB that can be repaired by nonhomologous end joining NHEJ or HDR (Figure 1). NHEJ can proceed either by the canonical or alternative pathways, resulting in the formation of insertions and/or deletions (indels) [67,68], leading to a loss of function or “knockout” of the target gene. On the other hand, HDR requires a donor DNA template with sequences homologous to the ends of DSBs. HDR allows for the precise replacement of the target gene, consequently generating a more precise edit. However, HDR has a lower efficiency than NHEJ, and various strategies, such as timed delivery of Cas9 ribonucleoprotein complexes [69] and delivery of a donor gRNA with a target region flanked by two sgRNA homology arms [70], have been explored to increase the rates of HDR.

The creation of DSBs in the Cas9 protein occurs through the HNH nuclease domain and the RuvC-like nuclease domain. The former cleaves the guide-bound target DNA strand, while the latter cleaves the PAM-containing complementary strand. Mutating the RuvC catalytic domain results in its inactivation, resulting in the cleavage of only one strand of DNA by the HNH domain [63]. This variant, termed nickase Cas9 (nCas9), has been used widely in base editing.

CRISPR base editing allows for the generation of site-specific point mutations without production of DSBs [71]. In general, CRISPR base editors consist of an nCas9 or catalytically inactive “dead” Cas protein, also known as dCas9, linked to a deaminase that can convert one target DNA base to another [71]. The Cas nuclease is directed by sgRNA to the target, where the formation of the Cas–sgRNA–DNA complex results in denaturation of the double helix, exposing a region of single-stranded DNA that can be modified by the linked deaminase (e.g., conversion of cytosine to uracil with cytosine base editors) [72,73]. The resulting base mismatch can then be repaired by cellular repair mechanisms (e.g., conversion of uracil to thymine post base editing).

Similarly, CRISPR activation (CRISPRa) and CRISPR interference (CRISPRi), known collectively as CRISPRmod, consist of dCas9 fused to a transcriptional effector. CRISPR interference (CRISPRi) involves fusing dCas9 to transcriptional repressor(s). Early work suggests that CRISPRi works by blocking transcription, being more efficient in bacteria as compared to mammalian cells [74]. Later work further improved the efficiency of CRISPRi in human cells by fusing dCas9 to known repressive chromatin-modifying domains such as the KRAB domain [75]. In addition, activating effectors such as VP16 can also be fused to dCas9, allowing for activation of the target gene [76]. CRISPRmod has mainly been used in human cell lines such as HEK293 [77]; however, more recent work has sought to expand its repertoire to cultured primary cells such as T cells [78] and hematopoietic stem cells [79].

CRISPRi has been utilised to silence *Pcsk9* as a proof of concept in vivo. Delivery of a dSaCas9-based repressor (dSaCas9^KRAB^) utilising a dual AAV8 system into the liver of adult mice resulted in significant decreases in Pcsk9 protein, as well as serum LDL levels, for up to 24 weeks [80].

## 4. Genome Editing Approaches for Coronary Artery Disease

Genome editing tools have been utilised to both understand and treat the drivers and consequences of CAD. In the following sections, the role of genome editing in these areas will be further elaborated on, and is summarised in Table 2.

### 4.1. PCSK9

#### 4.1.1. CRISPR-Based Editing of PCSK9

Numerous studies have pointed to the role of the *PCSK9* gene in maintaining cholesterol homeostasis. Additionally, loss-of-function *PCSK9* variants have been associated with a decrease in risk of coronary heart disease as well as extracoronary atherosclerotic phenotypes [31,116,117]. Consequently, many of the gene editing studies have focused on this gene. PCSK9 is mainly expressed in the liver, an easily targetable site by a variety of gene delivery vehicles such as adenoviral vectors and lipid nanoparticles.

In a 2014 study, Ding et al. explored the utility of CRISPR/Cas9 to disrupt mouse *Pcsk9* as a proof of principle. Utilising an adenoviral vector, they delivered the gene encoding SpCas9 and an sgRNA-targeting *Pcsk9* to C57BL/6 mice. Within a few days of the NHEJ-mediated *Pcsk9* disruption, they observed a ~35–40% decrease in plasma cholesterol levels in addition to a decrease in plasma PSCK9 levels and an increase in LDLR levels [96].

A later study utilised a liver-specific human *PCSK9* knock-in mouse model. In this hypercholesteraemic model, the CRISPR/Cas9-mediated disruption of human *PCSK9* resulted in reduced plasma levels of human PSCK9, but not mouse, and a concurrent decrease in plasma cholesterol levels. In the same study, the authors utilised third-generation base editor BE3 and sgRNAs to target both mouse and human PCSK9. Similarly, significant reductions in the levels of mouse and human PCSK9 were observed. Interestingly, mice injected with the BE3 constructs exhibited more precise gene disruptions as compared to mice injected with CRISPR/Cas9 constructs, with no off-target editing or chromosomal translocations observed [94]. These observations were in concordance with an earlier study similarly utilising an adenoviral vector to deliver BE3 and an sgRNA-targeting *Pcsk9* in a mouse model. Base editing resulted in a median rate of 25% base-edited alleles in the liver, while a low (~1%) of indels were detected. Similarly, the authors observed a greater than 50% reduction in plasma PCSK9 protein levels as well as a ~30% reduction in plasma cholesterol levels [110].

#### 4.1.2. Cas9 Variants

One potential reason for the use of the adenoviral vector is the large size of the gene encoding SpCas9 (~4.2 kb), which prevents the usage of the less immunogenic adeno-associated virus (AAV). Utilising a Cas9 from *Staphylococcus aureus* (SaCas9), Ran et al. were able to package both the SaCas9 gene and an sgRNA-targeting *Pcsk9* into a single AAV8 vector. Similarly, they observed a decrease in serum PCSK9 levels and a ~40% decrease in total cholesterol one week after administration, which was sustained till the end of the study at 4 weeks [100]. Other alternatives to SpCas9 also include *Neisseria meningitidis* Cas9 (NmeCas9). In a proof-of-concept study, Ibraheim et al. delivered NmeCas9 with an sgRNA-targeting *Pcsk9* in a single AAV vector, resulting in significantly decreased cholesterol levels 50 days after systematic administration of the vector. While a humoral response to NmeCas9 was elicited, the authors did not report signs of liver damage [98]. Furthermore, they observed that this Cas9 variant had minimal off-target edits, similar to their previous observations [118]. These SpCas9 alternatives not only expand the potential sequences that can be targeted with CRISPR, but also broaden the choice of vectors available for therapeutic purposes.

In addition to utilising CRISPR/Cas9 systems to target *PCSK9,* an engineered meganuclease delivered by an AAV8 vector has been used as well. This resulted in up to 60% reduction in serum PCSK9 levels and up to 84% reduction in serum LDL cholesterol levels with a first-generation engineered meganuclease. However, multiple off-target cleavages were detected with this first-generation meganuclease. A second-generation meganuclease was also generated in an attempt to reduce off-target editing [111]. A 3-year follow-up showed that on-target editing of liver PCSK9 was sustained with a stable reduction in serum PCSK9 and LDL cholesterol levels, highlighting the potential of this method of gene editing. However, concerns remain over AAV integration at the cut sites as well as the potential immunotoxicity of this approach [111,112]. More recently, reductions in off-target activity of the meganuclease were observed after delivery with a modified self-targeting vector and a shorter promoter [119].

#### 4.1.3. Nonviral Delivery Methods

CRISPR/Cas9 systems have also been delivered by nonviral vectors such as virus-like particles (VLPs) as well as lipid nanoparticles (LNPs). Banskota et al. utilised a VLP system to deliver an adenine base editor 8e (ABE8e) ribonucleoprotein (RNP) that can target and disrupt the *Pcsk9* gene in mice. Systematic administration of the VLP resulted in 63% editing in bulk liver, which was comparable to editing efficiencies obtained by CRISPR delivered by other delivery systems such as AAVs and LNPs, as well as a 78% decrease in serum Pcsk9 protein levels 1 week post injection. Furthermore, they observed that base editing was highly specific to the liver, and minimal off-target editing was observed above background in potential off-target loci identified by CIRCLE-seq, an in vitro screening method for identifying genome-wide off-target mutations of CRISPR/Cas9 [102]. While this approach is relatively new, it appears to hold great promise as a delivery platform.

LNPs have also been used as an alternative delivery platform. Rothgangl et al. utilised LNPs encapsulating an ABEmax mRNA and an sgRNA-targeting *Pcsk9* to disrupt the canonical *Pcsk9* splice site. They induced ~61% base editing of the *Pcsk9* gene in mice, resulting in a stable reduction of 95% plasma PCSK9 and 58% LDL cholesterol levels. Additionally, administration of this LNP construct in macaques resulted in ~26% base editing coupled with a 32% decrease in serum PCSK9 and a 14% reduction in LDL cholesterol levels. A subset of macaques was given a second dose 2 weeks later; however, no increase in editing rates was observed. Additionally, they detected antibodies against SpCas9 in this subgroup, indicating that there was a possibility of an ABE-specific T cell response. In general, off-target delivery of the LNP complex remained low, though the authors observed editing rates of 6–12% in the spleen. However, the study endpoint of 29 days was too short to fully understand the long-term safety of this approach [108].

In a separate study involving cynomolgus monkeys, Musunuru et al. similarly delivered a CRISPR adenine base editor with LNPs to introduce a single-nucleotide loss-of-function mutation in *PCSK9*. Unlike the Rothgangl et al. study, Musunuru et al. employed the ABE8.8 m base editor, which has increased editing efficiency as compared to older-generation ABE base editors [120]. Initial studies in mice showed that 1 week post intravenous administration of LNPs, there was ~70% base editing in the liver. Similarly, in cynomolgus monkeys, the liver remains the main editing site, though some *PCSK9* editing in the spleen remains. However, minimal off-target editing was detected in both monkey liver samples and primary cynomolgus monkey hepatocytes. In a longer-term study of up to 8 months, base editing frequencies were ~66%, with a stable reduction of 90% of serum PCKS9. Similarly, persistent reductions of LDL cholesterol (60%) and lipoprotein(a) (35%) were observed. Similar to the Rothgangl et al. study, moderate increases in blood transaminases attributed to LNP treatment were observed that resolved within two weeks [101].

Taken together, these studies highlight the various methods that different groups have taken to safely deliver and knock down PCSK9. LNPs appear to be a viable method of delivering genome editing technologies such as base editors to the liver, and, indeed, VERVE-101, a single-dose CRISPR base editing therapeutic targeting *PCSK9* has recently begun phase 1 b study in patients with heterozygous familial hypercholesterolemia and cardiovascular disease (NCT05398029). Early studies in nonhuman primates have shown good tolerance to the LNP-delivered therapeutic, with sustained reductions in blood PCSK9 and LDL cholesterol up to 476 days after dosing [106]. If similar successes can be observed in human trials, it has the potential to be a convenient one-dose therapeutic as compared to current monoclonal antibody-based treatments, which require more frequent dosing.

### 4.2. ANGPTL3

Angiopoietin-like 3 (ANGPTL3) is a protein primarily expressed in the liver that regulates lipoprotein levels in blood. It functions by inhibiting lipoprotein lipase, a key enzyme that is involved in the intravascular lipolysis of certain triglycerides. In homozygous, for a loss-of-function mutation in *Angptl3* resulting in the production of a truncated Angptl3, hypolipidemia was observed, associated with an increase in lipoprotein lipase activity and increased lipolysis of VLDL triglycerides [121]. Loss-of-function mutations in *ANGPTL3* have also been linked to lower blood triglycerides and LDL cholesterol levels, as well as a lower risk of CAD in human cohorts [122].

In line with this, Chadwick et al. utilised an adenoviral vector to deliver a cytosine-to-thymine base editor (BE3) with a guide RNA-targeting *Angptl3* to 5-week-old C57BL/6J mice aimed at generating nonsense mutations at codon Gln-135. Deep sequencing performed at the day 7 time point showed that the mice injected with the BE3-*Angptl3* construct had a median base editing rate of 35% with no evidence of off-target editing. Administration of this construct into a hyperlipidaemic *Ldlr*-knockout mice model that phenocopies patients with homozygous familial hypercholesterolemia resulted in a substantial (>50%) reduction in triglycerides and cholesterol in comparison with the BE3 control [109]. Considering that the 35% base editing rate already results in substantial reductions in cholesterol levels, it is likely that improvements in base editing efficiency can further improve the efficacy of this targeting method.

A recent study by Zuo et al. utilised a dual AAV construct to specifically deliver AncBE4max, a cytosine base editor, targeting mouse *Angptl3*, to generate a premature stop codon (Q135X) in the coding sequence of the gene. In this study, a higher base editing efficiency of ~63% was observed in the liver, with a near complete knockout of ANGPTL3 protein expression 2–4 weeks post AAV administration. Similarly, *Angptl3* base editing also resulted in >50% reductions in triglyceride and total cholesterol levels in serum [103].

More recently, Verve Therapeutics published preclinical data on the administration of an LNP-based adenine base editor targeting *ANGPTL3* aimed at disrupting the gene (VERVE-201) via generation of a premature stop codon. In a nonhuman primate (NHP) model of homozygous familial hypercholesterolemia and in wildtype monkeys, administration of the NHP homologue (VERVE-201cyno) resulted in reductions of ~88% in circulating ANGPTL3 90 days post administration [123]. Later publications showed that administering the murine surrogate (VERVE-201mu) in an *Ldlr* KO model of homozygous familial hypercholesterolemia led to a mean 47% decrease in LDL cholesterol after administration of VERVE-201, and an accompanying decrease in triglyceride concentrations in blood. Treatment of NHPs with VERVE-201cyno was also observed to induce mean liver *ANGPTL3* editing of up to 63%, with a 95% reduction in ANGPTL3 protein levels 7 days after receiving the higher dose of 3.0 mg/kg. More importantly, NHPs treated with the LNP construct did not exhibit any signs of liver damage after administration [124]. These results suggest that *ANGPTL3* base editing could be potentially safe and efficacious in future human trials.

In a separate study, a novel LNP was used to deliver Cas9 mRNA and an sgRNA-targeting *Angptl3* for knockdown of liver *Angptl3*. The 306-O12B LNP construct was delivered specifically to the liver, with a median editing rate of 38.5% and a 65.2% reduction in ANGPTL3 in serum. This delivery method did not result in detectable off-target editing or significant changes to blood transaminases (ALT and AST) levels. Specifically, the authors compared their LNP construct to the current FDA-approved construct of MC-3. Seven days post administration, 306-O12B LNP-mediated delivery of the Cas9 components resulted in higher editing rates as compared to MC-3-mediated delivery of the same, with serum ANGPTL3, LDL cholesterol, and triglyceride levels being lower in the 306-O12B LNP-treated mice. Taken together, results of this study support a compelling argument for more detailed preclinical studies utilising this unique LNP construct [81].

### 4.3. APOC3 and Other Lipid-Related Genes

#### 4.3.1. APOC3

Apolipoprotein C3 (*APOC3*) is another potential therapeutic target for coronary heart disease treatment. ApoC-III is an apolipoprotein that is synthesised mainly in the liver, and it has roles in inhibiting lipoprotein lipase-mediated lipolysis of triglyceride-rich lipoproteins [125]. Elevations in circulating ApoC-III levels have been correlated with increased triglyceride levels and an increase in risk for myocardial infarctions and CAD [126]. Loss-of-function mutations in *APOC3* have similarly been associated with lower levels of plasma triglycerides and a lower risk of CADs [127]. In a clinical trial evaluating an antisense oligonucleotide (ASO) targeted at *APOC3* mRNA, patients administered this ASO exhibited reduced apoC-III and triglycerides [128].

CRISPR/Cas9 systems have been used to generate APOC3 knockout rabbits [83], as well as hamsters [84]. In both animal models, *ApoC3* knockout results in the animals exhibiting fewer atherosclerotic lesions as well as a less atherogenic lipid profile on a high-fat diet. Similarly, ZFNs have been used to generate apoCIII knockout rabbits. Under a high-cholesterol diet, these rabbits similarly exhibited reduced atherosclerosis as well as lower total cholesterol levels compared to wild-type rabbits [44]. Recent work from Xu et al. on *ApoC3* deficiency in LDLR-deficient hamsters suggests that loss of ApoC3 may paradoxically accelerate the atherosclerotic phenotype observed in LDLR-deficient hamsters on a high-cholesterol diet [129]. Further investigation on the effects and benefits of APOC3 inhibition in different animal models of atherosclerosis need to be carried out.

#### 4.3.2. ApoE

Apolipoprotein (apo) E is an important protein in the suppression of atherosclerosis expressed not only in the liver but also in the brain and adrenal gland [130]. It is a key regulator of lipid levels in the body by way of mediating the clearance of lipoproteins from the circulation [131]. It is thus unsurprising that *ApoE* knockout mice have often been used in the study of atherosclerosis. Gene editing technologies have been applied to generate such animal models in recent years. For instance, TALEN-mediated gene editing was used to knockout *ApoE* in rats, resulting in a phenotype of dyslipidaemia after a high-cholesterol diet, which was not observed in wild-type rats. Furthermore, partial ligation of the carotid artery resulted in formation in atherosclerotic plaques in knockout mice after a high-cholesterol diet [113]. Similar observations were detected in Bama miniature pigs after CRISPR/Cas9-mediated knockout of *ApoE*. Pigs with frameshift mutations developed severe hypercholesterolemia and atherosclerotic lesions after 6 months of a high-fat, high-cholesterol diet, recapitulating a human-like phenotype [85]. Additionally, double knockouts of both *ApoE* and *LDLR* were also explored in Bama minipigs [86] and rabbits [91] to generate animal models of atherosclerosis.

#### 4.3.3. LDLR

The low-density lipoprotein receptor (LDLR) is a key regulator of cholesterol homeostasis responsible for the internalisation of cholesterol-laden lipoprotein particles [132]. Loss-of-function mutations in *LDLR* have been linked to increased LDL cholesterol levels in serum and atherosclerosis. Utilising CRISPR/Cas9 gene editing, *LDLR*-KO New Zealand white rabbits were generated to study the spontaneous development of hypercholesterolemia and atherosclerosis [90], which may be a useful model for human familial hypercholesterolemia.

While most animal models have been generated via CRISPR/Cas9-mediated germline genome editing, somatic gene editing can also be used to generate models of diseases. In 2017, Jarrett et al. utilised AAVs to deliver gRNAs, targeting *Ldlr* and *Apob*. Singular delivery of the gRNAs resulted in efficient reductions of >50% in mRNA expressions of both *Ldlr* and *Apob* in a *Cas9*-expressing transgenic mouse model. Expectedly, *Ldlr* disruption resulted in hypercholesterolemia and atherosclerosis. However, the concomitant disruption of *Apob* resulted in a lowering of blood cholesterol and atherosclerosis protection [82]. A later work similarly utilised AAVs to disrupt *Ldlr,* though instead of SpCas9, Jarrett et al. utilised SaCas9, a smaller Cas9 ortholog. SaCas9 and a guide RNA that targets *Ldlr* (AAV-CRISPR) were packaged into AAVs and intraperitoneally injected into wild-type C57BL/6J mice. Mice placed on a “standard Western diet” for 20 weeks were noted to exhibit severe hypercholesterolemia and develop atherosclerotic lesions. While mice injected with the AAV-CRISPR constructs had no detectable off-target edits, insertions of the AAV genome were detected at on-target cut sites, similar to the 2017 study [99]. While this may not be a concern for the purpose of model development, more attention must be placed on it when developing AAV-based genome editing therapeutics for gene therapy in humans.

Similarly, Zhao et al. utilised a dual AAV system to deliver SpCas9 (AAV-Cas9) and an sgRNA-Donor construct to neonatal mice harbouring a *Ldlr^E208X^* mutation. The E208X mutation in mice is equivalent to the E207X mutation in human *LDLR*, a nonsense point mutation found in a patient with familial hypercholesterolemia [133]. These mutant mice lack a functional LDLR protein; thus, after 12 weeks of high-fat diet feeding, mutant mice had atherosclerotic lesions present in the aorta, unlike WT mice. These mice also exhibited higher total plasma cholesterol, recapitulating many of the features of familial hypercholesterolemia in humans. Post AAV-CRISPR/Cas9 treatment, these mice exhibited partial restoration of LDLR levels (~18%). Additionally, indels were detected in ~24% of the *Ldlr* alleles, and the HDR-mediated correction was observed in 6.7% of the *LDLR* alleles. Mice treated with the AAV-CRISPR/Cas9 system exhibited lower plasma cholesterol, triglycerides, and LDL-C, and a less severe atherosclerotic phenotype, suggesting that this could be a potential therapeutic approach for treatment of familial hypercholesterolemia [89].

While classical studies using *Ldlr*-KO mice cannot be replaced, somatic gene editing can allow for the simultaneous targeting of multiple genes, as well as avoid the issues of embryonically lethal knockouts. One can imagine that with this method, we can better understand the interactions of different genes on a disease background.

### 4.4. Genome Editing in the Understanding of the Complications Arising from CAD

Genome editing can also assist in the understanding the complications arising from CAD, such as myocardial infarction. Post MI, various complications can arise, such as ischemia-perfusion injury. Understanding the mechanisms and cellular consequences of myocardial infarction could provide new insights into potential strategies to minimise the damaging effects of the disease progression. In the following sections, we will briefly cover some of the potential genome editing targets that may ameliorate the impact of CADs (Figure 2). 

#### 4.4.1. CCC Complex

The COMMD/CCDC22/CCDC93 (CCC) complex is a large protein complex comprising CCDC22, CCDC93, and the COMMD proteins. In macrophages, the CCC complex has been found to be responsible for the recycling of surface proteins and has been found to associate closely with a cargo recognition complex called retriever [134]. The CCC complex is crucial for the function and localisation of the low-density lipoprotein receptor (LDLR). Ablation in *Commd1* expression results in increases in plasma LDL cholesterol levels as well as a mislocalisation of LDLR. [135]. CRISPR/Cas9-mediated deletion of *Ccdc22* in mouse livers resulted in decreased expression of all COMMD proteins except COMMD6, as well as elevation in plasma LDL cholesterol levels. Taken together, these indicate the importance of the CCC complex in the regulation of plasma cholesterol levels in a mouse model [87].

#### 4.4.2. HIF1

One of the major and early responses to myocardial infarctions by the body is an increase in the hypoxia-inducible factor 1 (HIF-1) transcriptional factor [136], where close to 200 genes are transcriptionally activated [137]. In the presence of hypoxia, HIF-1 enhances triglyceride synthesis and lipid droplet formation in foam cells. Additionally, it promotes lipid retention by upregulation of the expression of scavenger receptors such as CD36 and LOX1, and downregulation of the lipid exporter ABCA1. Additionally, HIF-1 can promote the proliferation and migration of smooth muscle cells, leading to plaque progression. Geng et al. discovered that UCHL1 is upregulated in the infarct and border zones of hearts following myocardial infarction. Utilising a birA-based proximity labelling system, HIF-1a, part of the HIF-1 complex, was discovered to interact with UCHL1. To investigate the function of UCHL1 in the regulation of HIF-1a, CRISPR/Cas9 genome editing was used to establish a UCHL^−/−^ hiPSC line. After hypoxia treatment, UCHL1^−/−^ hiPSC-cardiomyocytes exhibited lower levels of HIF-1a and its target genes as well as mislocalisation of HIF-1a to the cytosol. Furthermore, KO of UCHL1 resulted in enhanced ubiquitination of HIF-1a. Taken together, these suggest that the localisation and stability of HIF-1a is linked to UCHL1 [138].

#### 4.4.3. Alpha-Arrestins

Alpha-arrestins have been known to respond to glucose availability in yeast [139], and recently, human homologs have been studied to better understand their function. ARRDC4 is a mammalian alpha-arrestin whose role in vivo has yet to be clearly understood. Utilising CRISPR/Cas9, Nakayama et al. generated a novel Arrdc4-KO mouse model aimed at better understanding the role of ARRDC4 in vivo. Ultimately, they discovered a new mechanism in which ARRDC4 plays a role in glucose deprivation and endoplasmic reticulum stress during ischemia, which may provide a potential therapeutic target for the ischemic heart [140].

#### 4.4.4. Atherosclerosis and Monocyte Activation

In a 2022 study, Karamanavi et al. utilised CRISPR genome editing to study the impact of the rs17514846 and rs1894401 SNPs on *FES* (FES proto-oncogene, tyrosine kinase) expression in monocytes. FES is postulated to have a protective role against atherosclerosis. The two SNPs of interest are located near *FURIN* and *FES* and have been shown to have an expression quantitative trait loci effect on *FES* expression in atherogenic cell types of monocytes and vascular smooth muscle cells. THP-1 monocytes were edited with CRISPR to generate isogenic cells that only differ in either of the two SNPs. Cells containing the CAD risk genotype of rs17514846 A/A exhibited lower FES levels than cells containing the C/C genotype, while cells with the rs1894401 G/G genotype, which has been found to be in high linkage disequilibrium with rs17514846 A/A genotype, similarly exhibited lower FES levels. These suggest that both SNPs have a role in modulating FES expression in monocytes [141]. However, it is still unclear if base editing of the rs17514846 and rs1894401 SNPs to a nonrisk genotype has a beneficial effect on FES activity and, by extension, atherosclerosis.

#### 4.4.5. MIA3

GWAS studies have recently identified rs67180937 at the 1q41 locus as an SNP associated with lower VSMC proliferation and MIA SH3 Domain ER Export Factor 3 (MIA3) expression. MIA3, also known as TANGO1, is a ubiquitously expressed protein that is involved in the export of collagen from the endoplasmic reticulum as well as angiogenesis and leukocyte migration, a phenomenon that has been linked to atherosclerosis [142]. While preliminary data have suggested that lower MIA3 may be linked to the formation of a thin fibrous plaque cap resulting from lower VSMC proliferation, it is still not understood what role MIA3 plays [143]. It may be possible to utilise base editing on a mouse model of atherosclerosis to insert the rs67180937 SNP and examine its impact on the atherosclerotic phenotype.

#### 4.4.6. CaMKIIδ

Calcium calmodulin-dependent protein kinase IIδ (CaMKIIδ) is multifunctional Ser/Thr kinase, a member of the CaMKII family. CaMKIIδ has been found to be the predominant member of the CaMKII family expressed in the heart, and has been linked to mediating various signalling processes in cardiomyocytes [144]. Chronic overactivation has been linked to various cardiac diseases such as ischemia/reperfusion injury and heart failure. In particular, two methionine residues on the regulatory domain of CaMKIIδ have been identified to promote hyperactivation of the kinase. In a 2023 study by Lebek et al., the Abe8e base editor fused to a Cas9 variant, SpRY, was used to ablate the oxidation of the methionine residues. Delivery of the AAV9 constructs expressing the CRISPR/Cas9 gene editor immediately post cardiac ischemia/reperfusion injury resulted in improvements in cardiac function [104]. Results from this study suggest that CaMKIIδ gene editing can indeed be a viable therapeutic approach for heart disease.

#### 4.4.7. TLR4

Interestingly, CRISPR/Cas9 editing has also been used to disrupt toll-like receptor 4 (TLR4) to improve the outcome of cell therapy for CVDs. TLR4 is a regulator of inflammation and is expressed in many cell types, including cardiomyocytes. Activation of TLR4 leads to the activation of several key transcription factors such as NFκB, AP-1, and IRF3, which promote the expression of proinflammatory cytokines and interferons that promote plaque progression. Cardiac ischemia/reperfusion often results in cellular damage, which in turns results in the activation of TLR4 and proinflammatory responses [145]. Human mesenchymal stromal cells (hMSCs) have been cited as a viable form of cardiac cell therapy; however, it has been hypothesised that the environment of the diseased heart could drive these hMSCs to a proinflammatory phenotype mediated by TLR4. Schary et al. utilised CRISPR/Cas9 to edit hMSCs to lower TLR4 expression. Mice administered with the edited hMSCs post MI exhibited improved survival, infarct healing, and remodelling [146]. While results of the study are promising, concerns regarding immune responses to hMSC implantation at the site of injury remain. Furthermore, more work must be carried out in generating a more heterogenous population of edited and unedited cells; however, this approach appears to be promising for translation into human therapy.

Taken together, these studies show a role of genome editing technologies in not only understanding the mechanisms driving the deleterious effects of MI, but also highlighting potential ways we can target them therapeutically. However, more work in larger animal models must be carried out to assess the safety and efficacy of these therapeutic approaches before translation to human therapy is possible.

## 5. Potential Challenges Facing Genome-Editing-Based Therapies

While genome editing is indeed a promising therapeutic approach, there are still concerns regarding its potential side effects. Undesirable side effects of ZFNs include insertional mutagenesis, toxicity, and low efficacy [147]. Off-target effects are lower in CRISPR/Cas9 technology compared to ZFNs, though prevalence of unpredictable off-target effects, which include off-target mutations, can still be high (up to >50%). CRISPR-induced DSBs can cause apoptosis, leading to DNA damage and cellular toxicity. Off-target effects are far rarer in TALEN-mediated systems, and their high degree of specificity and low cytotoxicity have been shown in diverse cell types, though it comes at the cost of them being more expensive and labour-intensive than their fellow genome editing counterpart, CRISPR/Cas9 [148]. Often, off-target effects for the CRISPR/Cas9 system can results from a mismatch between the guide RNA and double-stranded DNA [149,150]. As alluded to in the preceding sections, off-target editing remains a concern. Many bioinformatic tools have been developed to design gRNAs as well as to detect potential off-targets (e.g., CRISPRseek [151], COSMID [152]) to overcome this challenge. Other strategies to reduce off-target modification by CRISPR/Cas9 include the use of Cas9 nickase, which breaks down only one strand of the DNA and thereby reduces further damage of the target DNA, and anti-CRISPR proteins (ie. Acr), which deactivate the Cas9 protein after targeting its site [153]. The risk of off-target effects of CRISPR/dCas9 is much lower than Cas9, and tools such as CRISPRa and CRISPRi that use dCas9 are reversible, which reduces the number of unknown problems associated with off-target effects [150]. Additionally, depending on the choice of delivery vector, there are concerns surrounding potential integration of vector DNA into the genome. Other potential issues that may arise from such therapies are their persistence in the tissue of interest. While work in animal models has shown that effects of genome editing can persist for up to 476 days after dosing [106], it is unclear if similar observations will be made in humans.

## 6. Conclusions

Genome editing technologies have indeed been greatly utilised over the years not only in the development of disease models, both in animals and cells, but also in the treatment of disease. Various methods ranging from nonviral lipid nanoparticles to adenoviral vectors have been explored to deliver these gene editors to animal models. However, questions of immunogenicity and efficacy of gene editing remain. Currently, lipid nanoparticles appear to be one of the safer means of delivering gene editors in vivo with minimal risk of unwanted integration into the genome. Additionally, gene editing of PCKS9 and ANGPTL3 appears to be a promising avenue of therapy, with two LNP-delivered therapeutics already in preclinical study. It will be exciting to see if this approach will prove to be efficacious in human trials.

## Figures and Tables

**Figure 1 ijms-24-14057-f001:**
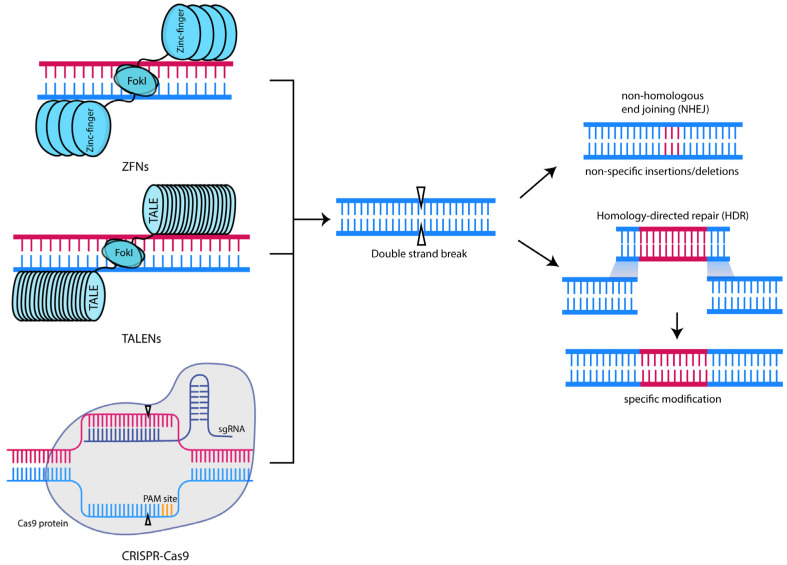
Genome editing technologies and their mechanism of action. Except for base editors, most of the major genome editing technologies utilised today function by recognising their target site and inducing a double-stranded break. These DSBs can be repaired by either NHEJ or, if a donor DNA template is provided, HDR.

**Figure 2 ijms-24-14057-f002:**
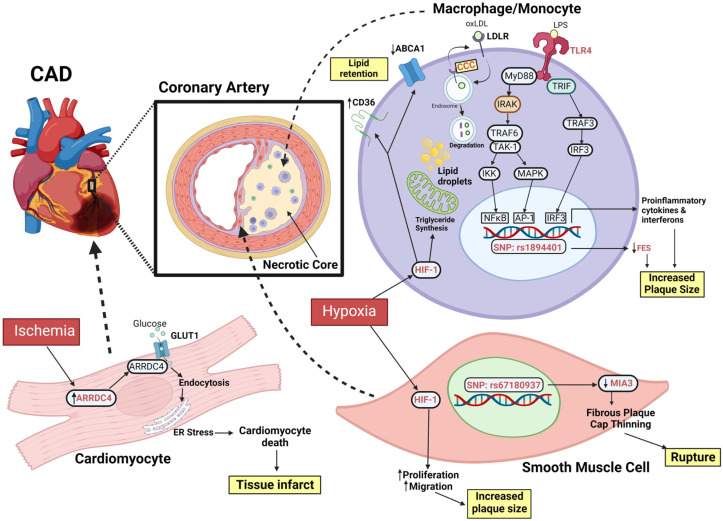
Schematic of key molecular targets for genome editing in the complications arising from CAD. In summary, the three groups of cells involved are monocytes/macrophages, smooth muscle cells, and cardiomyocytes. In macrophages, the CCC complex plays an important role in the recycling of surface proteins such as LDLR. TLR4 plays an important role in the proinflammatory response. Its activation results in expression of key transcription factors that promotes the expression of proinflammatory cytokines and interferons that promote plaque progression. In the presence of hypoxia, HIF-1 enhances triglyceride synthesis and lipid droplet formation in foam cells and promotes lipid retention. FES has been shown to have a protective role against atherosclerosis, and the rs1894401 and rs1751486 SNPs (latter not shown) located at the same locus both decreased FES expression. In the ischemic heart, the alpha arrestin ARRDC4 is upregulated and plays an important role in interacting with GLUT1 and causing ER stress, and subsequently promoting cardiomyocyte death. Abbreviations: ABCA1, ATP Binding Cassette Subfamily A Member 1; AP-1, activator protein 1; ARRDC4, arrestin domain containing 4; CAD, coronary artery disease; CCC, COMMD-CCDC-CCDC93 complex; ER, endoplasmic reticulum; FES, FES proto-oncogene; HIF-1, hypoxia-inducible factor-1; GLUT1, glucose transporter 1; IKK, IκB kinase; IRAK, interleukin-1 receptor associated kinase; IRF3, interferon regulatory factor 3; LPS, lipopolysaccharide; MAPK, mitogen-activated protein kinase; MIA3, MIA SH3 domain ER export factor 3; NFκB, nuclear factor kappa light chain enhancer of activated B cells; oxLDL, oxidated low-density lipoprotein; SNP, single nucleotide polymorphism; TAK-1, transforming growth factor-activated kinase 1; TLR4, toll-like receptor 4; TRAF3/6, TNF receptor associated factor 3/6; TRIF, TIR-domain-containing adapter protein inducing interferon beta; created with BioRender.com.

**Table 1 ijms-24-14057-t001:** Summary of genome editing approaches.

Technology	General Mechanism	Strengths	Weaknesses
ZFNs	Zinc finger domains conjugated to a *Fok*I endonuclease domain recognises target sequence. Dimerisation of *Fok*I domain allowed for activity and cleavage of target sequence. Each zinc finger binding domain recognises 3 nucleotides. Recognition sites are usually at least 18 base pairs long.	Easily delivered using viral and nonviral delivery vectors.	Protein engineering required to generate different ZFNs. Not as specific compared to TALENs.
TALENs	TALEs conjugated to a *Fok*I endonuclease domain recognises target sequence. Dimerisation of *Fok*I domain allowed for activity and cleavage of target sequence. Each TAL repeat unit recognises 1 nucleotide. Recognition sites are usually at least 14 base pairs long.	More specific compared to ZFNs. Can target heterochromatin with greater efficiency compared to CRISPR/Cas systems. Easier to engineer for specific sequence targeting as compared to ZFNs	Hard to deliver with viral vectors due to large size of TALENs. Possibility of off-target editing remains.
CRISPR/Cas	Guide RNAs direct Cas protein to the target site for cleavage. Guide sequences are generally 20 base pairs long.	Easy to multiplex. Easily cloned and synthesised. Easy to modify to target novel sequences as long as a PAM site is present.	Typically used spCas9 protein is relatively large, difficult to deliver with viral vectors (e.g., AAVs), but possible to use smaller orthologs. Possibility of off-target editing.

**Table 2 ijms-24-14057-t002:** Summary of various applications of genome editing technologies in vivo.

Gene Editor	Delivery Vector	Target Gene	Model	Model Phenotype	Reference
CRISPR/Cas9	LNPs	*Angptl3*	6–80-week-old female C57BL/6 mice	65.2% reduction in serum ANGPTL3 protein, lower LDL cholesterol and triglyceride levels.	[81]
CRISPR/Cas9	AAV8	*Apob*	6–9-week old Cas9 transgenic male mice	Mice with both *Ldlr* and *Apob* KO showed rapid drop in plasma cholesterol. Near complete loss of both Apob and LDLR protein. No atherosclerotic lesions.	[82]
CRISPR/Cas9	Microinjection and embryo transfer	*Apoc3*	New Zealand White rabbits	Chow diet: KO rabbits had 50% lower triglyceride levels and increased plasma lipoprotein lipase levels. High fat diet: No change in plasma triglycerides, total cholesterol, and LDL cholesterol levels. Mild atherosclerotic lesions.	[83]
CRISPR/Cas9	not mentioned	*Apoc3*	Syrian golden hamsters	Reduced triglyceride and total cholesterol in blood, marked increase in HDL cholesterol. Fewer atherosclerotic lesions in both thoracic and abdominal arteries compared to WT.	[84]
CRISPR/Cas9	Somatic Cell Nuclear Transfer (SCNT)	*ApoE*	Bama miniature pigs	Normal diet: KO pigs have moderately increased plasma cholesterol levels. High-fat, high-cholesterol diet: severe hypercholesterolemia, human-like atherosclerotic lesions in aorta and coronary arteries.	[85]
CRISPR/Cas9	SCNT	*ApoE*, *LDLR*	Bama minipigs	Significant elevation in LDL cholesterol, total cholesterol, and apolipoprotein B.	[86]
CRISPR/Cas9	Adenovirus	*Ccdc22*	Liver-specific Cas9-expressing C57BL/6J mice	70% reduction in CCDC22 levels, decreased expression of all COMMD proteins except COMMD6, ~35% increased plasma total cholesterol.	[87]
CRISPR/Cas9	mRNA	*LCAT*	Golden Syrian hamster	Extremely low HDL in plasma, hypertriglyceridemia. Proatherogenic dyslipidaemia.	[88]
CRISPR/Cas9	AAV8	*Ldlr*	P1 and P2 *Ldlr^E208X^* mutant mice	Significant reduction in total cholesterol, triglycerides, and LDL cholesterol in serum. Smaller atherosclerotic plaques in aorta.	[89]
CRISPR/Cas9	AAV8	*Ldlr*	6–9-week-old Cas9 transgenic male mice	Significantly higher plasma cholesterol. Visible atherosclerotic lesions. Near complete loss in LDLR protein expression.	[82]
CRISPR/Cas9	microinjection of zygote	*LDLR*	New Zealand White rabbits	Spontaneous development of hypercholesterolemia and atherosclerosis on normal chow diet.	[90]
CRISPR/Cas9	Microinjection and embryo transfer	*LDLR*, *apoE*	New Zealand White rabbits	Normal chow diet: 10× higher cholesterol levels compared to WT, aortic, and coronary atherosclerosis.	[91]
CRISPR/Cas9	AAV8	*Ldlr^E208X^*	Not stated	Severe atherosclerotic phenotype after high-fat diet.	[89]
CRISPR/Cas9	Magnetoplexes	*miR34a*	8-week-old ICR mice that underwent MI	Decreased miR34a expression, reduced collagen fibril formation, increased proliferation and cardiomyocyte number.	[92]
CRISPR/Cas9	Adenovirus	*Pcsk9*	FRG KO mouse model, engrafted with primary human hepatocytes	52% reduction in human PCSK9 levels. Twofold increase in mouse Pcsk9 levels. No change in total cholesterol levels.	[93]
CRISPR/Cas9	Adenovirus	*Pcsk9*	28-week-old hPCSK9-KI mouse model	Decreased Pcsk9 protein and mRNA levels in liver. Significant reduction in total plasma cholesterol and LDL cholesterol levels.	[94]
CRISPR/Cas9	AAV8	*Pcsk9*	4- to 6-week-old C57/BL6J male mice	~80% decrease in serum Pcsk9. ~35% decrease in total cholesterol after 24 weeks.	[95]
CRISPR/Cas9	Adenovirus	*Pcsk9*	5-week-old female C57BL/6 mice	35–40% reduction in total plasma cholesterol. Substantially lower Pcsk9 levels.	[96]
CRISPR/Cas9	gold nano clusters	*Pcsk9*	6 weeks old C57BL6/J female mice	Decrease in serum LDL cholesterol, reduced serum Pcsk9 level.	[97]
NmeCas9	AAV8	*Pcsk9*	12- to 16-week-old female C57BL/6 mice	Lower cholesterol levels. Decreased Pcsk9 levels.	[98]
SaCas9	AAV8	*Ldlr*	6 weeks old C57BL6/J (both sexes)	Decrease in LDLR protein expression. Hypercholesterolemia. Atherosclerotic lesion formation (more significant for male mice).	[99]
SaCas9	AAV8	*Pcsk9*	5–6-week-old male C57/BL6 mice	~95% decrease in serum Pcsk9. ~40% decrease in total cholesterol one week post administration, sustained throughout four-week course,	[100]
ABE8.8m base editor	LNPs	*Pcsk9*	Cynomolgus monkeys	81% reduction in serum Pcsk9. 65% reduction in serum LDL cholesterol.	[101]
ABE8e base editor	Virus like particles	*Pcsk9*	6- to 7-week-old adult C57BL/6J mice	~78% reduction in serum Pcsk9 levels 1 week post injection.	[102]
AncBE4max	AAV9	*Angptl3*	6-week-old B6 mice	~58% decrease in serum triglyceride levels. ~61% decrease in total cholesterol. ~88% decrease in ANGPTL3 protein.	[103]
CRISPR/Cas9 ABE	AAV9	*CaMKIIδ*	12-week-old male C57Bl6 mice	Similar levels of fractional shortening comparable to sham. LV end-diastolic dilation not observed.	[104]
CRISPR/Cas9 ABE	AAV8	*Pcsk9*	8-week-old female mice	~50% decrease in Pcsk9 protein level, lower VLDL/LDL levels 6–8 weeks post injection.	[105]
CRISPR/Cas9 ABE	LNPs	*Pcsk9*	Cynomolgus monkey	Up to 83% lower blood Pcsk9 protein and 69% lower low-density lipoprotein cholesterol up to 476 days after dosing.	[106]
CRISPR/Cas9 ABE	Lipid-like nanomaterials	*Pcsk9*	Balb/c mice	Reduction in serum Pcsk9 levels.	[107]
CRISPR/Cas9 ABE	AAV8	*Pcsk9*	5-week-old male C57BL/6J mice	Decrease in plasma Pcsk9 and LDL levels.	[108]
CRISPR/Cas9 ABE	LNPs	*Pcsk9*	Male C57BL/6J mice	95% reduction in plasma Pcsk9. 58% reduction in LDL cholesterol.	[108]
CRISPR/Cas9 ABE	LNPs	*Pcsk9*	Cynomolgus macaques	32% reduction in plasma Pcsk9. 14% reduction in LDL cholesterol.	[108]
CRISPR/Cas9 BE3	Adenovirus	*Angptl3*	5-week-old C57BL/6J male mice	49% reduction in plasma ANGPTL3. 31% reduction in plasma triglycerides. 19% reduction in cholesterol after 7 days.	[109]
CRISPR/Cas9 BE3	Adenovirus	*Angptl3*	5-week-old male hyperlipidaemic *Ldlr* KO mice	56% reduction in triglycerides. 51% reduction in cholesterol after 14 days.	[109]
CRISPR/Cas9 BE3	Adenovirus	*Pcsk9*	5-week-old C57BL/6J mice	>50% reduction in Pcsk9 protein levels. ~30% reduction in plasma cholesterol levels.	[110]
CRISPR/Cas9 BE3	Adenovirus	*Pcsk9*	10-week-old hPCSK9-KI mouse model	Significant reductions in levels of circulating human and mouse Pcsk9 protein. Significant reductions in plasma total cholesterol.	[94]
dSaCas9KRA	AAV8	*Pcsk9*	6–8-week-old C57Bl/6 male mice	Reduction in serum Pcsk9 and cholesterol levels.	[80]
I-CreI-based meganuclease first generation (M1PCSK9)	AAV8	*Pcsk9*	Nonhuman primate. Rhesus macaques	Up to 84% decrease in serum Pcsk9, up to 60% decrease in serum LDL.	[111]
I-CreI-based meganuclease second generation (M2PCSK9)	AAV8	*Pcsk9*	Nonhuman primate. Rhesus macaques	~63% reduction in Pcsk9 protein. ~40% reduction in serum LDL levels.	[112]
M2PCSK9	AAV8	*Pcsk9*	Nonhuman primate. Rhesus macaques	Up to 62% reduction in serum Pcsk9, up to 39% reduction in serum LDL.	[111]
TALEN		*ApoE*	6–8-week-old male rats	High cholesterol diet: typical dyslipidaemia, not observed in WT. No obvious atherosclerotic lesions on aorta and aortic root. Partial ligation of carotid arteries resulted in formation of plaques in KO rats.	[113]
TALEN	microinjection and embryo transfer	*human ApoAII*	New Zealand White rabbits	Knock-in rabbits had lower atherosclerotic burden, lower plasma triglycerides, and higher plasma HDL levels.	[114]
ZFN	Microinjection	*Apoc3*	Embryos	Normal diet: KO rabbits had significantly lower plasma levels of triglycerides but unchanged levels of total cholesterol and HDL (compared to WT). Cholesterol-rich diet: KO rabbits had significantly lower levels of plasma total cholesterol and triglycerides than WT.	[44]
ZFN	Lipid nanoparticle	*Pcsk9*	8- to 10-week-old C57BL/6 mice	>90% reduction in Pcsk9protein levels in plasma.	[115]

## Data Availability

Not applicable.

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
