# Peer review of "Applications of Genome Editing Technologies in CAD Research and Therapy with a Focus on Atherosclerosis"

_ijms, 2023, doi:10.3390/ijms241814057_

Round 1

Author Response

  1. What possible side effects may appear when applying these therapies. They are only mentioned in some (example, line 287)

Thank you very much for the comment. We have included a section (Section 5) to address these issues of side effects and off-target effects.

  1. In the same way, what can be the side effects in each of these therapies, if they are known. Again, they are only mentioned in a few.

Thank you very much for this comment. We have included in Section 5 the off-target effects of each of the therapies and how they compare with each other.

  1. Can you mention the possible effects in the long term or in longer duration therapies?

Thank you very much for this comment. We have included in Section 5 the potential long-term effects of genome editing in animal models, though information in this area in the literature is limited.

  1. Is it known if after a while of therapy, there is a reversal of the effect?

Thank you very much for this comment. We have included in Section 5 a few ways in which genome editing through CRISPR can be improved in terms of lowering off-target effects – with one of them being using dCas9 in CRISPRi or CRISPRa. As their effects are reversible, risks of off-target effects are also lower. Additionally, we have mentioned how off-target mutations could lead to target DNA damage and cell death, particularly in the case of CRISPR-Cas9, which could be irreversible.

Reviewer 2 Report

This manuscript summarize the application of gene editing in CAD. Allover, the logic is good but some concerns should be addressed.

.

Major concerns:

1.     In Figure 1, the diagram of CRISPR-Cas9 is not correct. The PAM site should not be included in 20nt spacer RNA. Please re-draw this diagram.

2.     When the abbreviation first shows up, you should spell out full name. For example, what does SCNT mean in Table 2. If you define the abbreviation before, please use it accordingly. For example, “Adenine base editing” can changed into “ABE”.

3.     For Table 2, you should re-organize it according to the gene editing technology, like ZNFs,TALEN,CRISPR.

4.     A diagram or table will welcome for the description of “Genome editing in the understanding of the consequences of CAD”

Minor concerns:

1.     In line 56 of page 2, “CRISPR can assist in furthering the understanding of and 56

treatment for CAD” changed into “CRISPR can assist in the further understanding of and treatment for CAD”;

2.     In line 127 of page 4, “Barring base editors” changed into “except base editors”;

3.     In line 475 of page 14, “Genome editing in the understanding of the consequences of CAD” changed into “Genome editing in the understanding of CAD”

Please carefully read and check the grammar of the manuscript.

English should moderately revised.

Author Response

Major concerns:

  1. In Figure 1, the diagram of CRISPR-Cas9 is not correct. The PAM site should not be included in 20nt spacer RNA. Please re-draw this diagram.

Thank you very much for this comment. We have re-drawn and amended this diagram accordingly.

  1. When the abbreviation first shows up, you should spell out full name. For example, what does SCNT mean in Table 2. If you define the abbreviation before, please use it accordingly. For example, “Adenine base editing” can changed into “ABE”.

Thank you very much for this comment. We have made the appropriate changes pertaining to the abbreviations throughout the tables and the manuscript.

  1. For Table 2, you should re-organize it according to the gene editing technology, like ZNFs,TALEN,

Thank you very much for this comment. We have re-organized Table 2 according to the gene editing technology as highlighted.

  1. A diagram or table will welcome for the description of “Genome editing in the understanding of the consequences of CAD”

Thank you very much for this comment. We have added a diagram (Figure 2) highlighting molecular targets and pathways associated with genome editing in the understanding of the complications arising from CAD. We have included a macro-perspective of the organ and tissue level, and highlighted important cell signaling pathways that play crucial roles during CAD and events that follow afterwards such as myocardial ischemia and infarction.

Minor concerns:

  1. In line 56 of page 2, “CRISPR can assist in furthering the understanding of and treatment for CAD” changed into “CRISPR can assist in the further understanding of and treatment for CAD”

Thank you very much for this comment. We have amended this line as suggested.

  1. In line 127 of page 4, “Barring base editors” changed into “except base editors”;

Thank you very much for this comment. We have made the change as recommended.

  1. In line 475 of page 14, “Genome editing in the understanding of the consequences of CAD” changed into “Genome editing in the understanding of CAD”

Thank you very much for this comment. We have amended this line with the intention to focus on complications that arise from CAD.

Please carefully read and check the grammar of the manuscript.

Thank you very much for this comment. We have read and checked the grammar of the manuscript as recommended.

Reviewer 3 Report

A nice  review on  applications of genome editing technologies in CAD  is  presented.  Several minor  issues could  be  clarified

1.       Most available  studies are limited to lipid management and  atherosclerosis. Thus  the  title  could  be   more specific

2.       Abstract should  give  more  information about  the  text body

English is fine.

Author Response

  1. Most availablestudies are limited to lipid management and   Thus  the  title  could  be   more specific.

Thank you very much for this comment. We have amended the title to be more specific as highlighted.

  1. Abstract shouldgive  more  information about  the  text body.

Thank you very much for this comment. We have added more information to the abstract about the main text body as recommended.
